# Intra-Host Evolution During Relapsing Parvovirus B19 Infection in Immunocompromised Patients

**DOI:** 10.3390/v17081034

**Published:** 2025-07-23

**Authors:** Anne Russcher, Yassene Mohammed, Margriet E. M. Kraakman, Xavier Chow, Stijn T. Kok, Eric C. J. Claas, Manfred Wuhrer, Ann C. T. M. Vossen, Aloys C. M. Kroes, Jutte J. C. de Vries

**Affiliations:** 1Medical Microbiology and Infection Control, Leiden University Center for Infectious Diseases (LUCID), Leiden University Medical Center (LUMC), 2333 ZA Leiden, The Netherlands; 2Center for Proteomics and Metabolomics, Leiden University Medical Center (LUMC), 2333 ZA Leiden, The Netherlands

**Keywords:** parvovirus B19, transplantation, humoral immune selection, whole-genome sequencing, protein modeling

## Abstract

Background: Parvovirus B19 (B19V) can cause severe relapsing episodes of pure red cell aplasia in immunocompromised individuals, which are commonly treated with intravenous immunoglobulins (IVIGs). Few data are available on B19V intra-host evolution and the role of humoral immune selection. Here, we report the dynamics of genomic mutations and subsequent protein changes during relapsing infection. Methods: Longitudinal plasma samples from immunocompromised patients with relapsing B19V infection in the period 2011–2019 were analyzed using whole-genome sequencing to evaluate intra-host evolution. The impact of mutations on the 3D viral protein structure was predicted by deep neural network modeling. Results: Of the three immunocompromised patients with relapsing infections for 3 to 9 months, one patient developed two consecutive nonsynonymous mutations in the VP1/2 region: T372S/T145S and Q422L/Q195L. The first mutation was detected in multiple B19V IgG-seropositive follow-up samples and resolved after IgG seroreversion. Computational prediction of the VP1 3D structure of this mutant showed a conformational change in the proximity of the antibody binding domain. No conformational changes were predicted for the other mutations detected. Discussion: Analysis of relapsing B19V infections showed mutational changes occurring over time. Resulting amino acid changes were predicted to lead to a conformational capsid protein change in an IgG-seropositive patient. The impact of humoral response and IVIG treatment on B19V infections should be further investigated to understand viral evolution and potential immune escape.

## 1. Introduction

Parvovirus B19 (B19V) is a 5·6 kb, single-stranded DNA virus. The viral genome codes for three major proteins: the viral capsid consists of a VP1 and VP2 protein, which share part of their coding sequences, while the non-structural (NS) protein is involved in replication and cellular processes [1]. Most infections occur in childhood, and their presentation in children with fever and rash is known as erythema infectiosum or ‘fifth disease’, a mild self-limiting disease. However, B19V may cause severe anemia in certain susceptible populations. B19V infects erytroid progenitor cells in the bone marrow, which leads to apoptosis and a halt in erythropoiesis. In healthy hosts, this halt is temporary. In severely immunocompromised individuals such as stem cell or solid organ transplantation patients, this halt is prolonged because of the absence of an effective immune response and may then lead to the clinical syndrome of ‘pure red cell aplasia’ (PRCA). PRCA can be treated by administering intravenous immunoglobulins (IVIGs), which usually leads to a reduction of the viral load and the alleviation of symptoms [2]. Because the effect of IVIG disappears with a decreasing IVIG titer, a pattern of recurrent infections may occur, each episode managed by IVIG until the patient’s own immunity is restored [3].

In immunocompromised individuals, the inability to clear common self-limiting infections can lead to the long-term persistence of viruses, creating a reservoir in which mutants may arise. Immune escape mutants have been observed during prolonged infections with a variety of viral pathogens, such as SARS-CoV-2 and influenza virus, as well as latent DNA viruses, including cytomegalovirus (CMV), during antiviral treatment [4,5,6,7]. In B19V treatment, IVIG suppresses replication only temporarily; replication flares after IVIG is cleared from the circulation (in the absence of a sufficient natural immune response). Due to the relapsing nature of B19V in immunocompromised individuals, we hypothesize that new viral variants may also emerge in B19V infections over time.

Previous studies showed some genetic drift in prolonged B19V infection in patients treated with IVIG; small numbers of point mutations have been reported in single case reports or small case series [8,9,10,11]. These studies used conventional Sanger sequencing for analysis and mostly looked at partial genomes, focusing on capsid proteins. Over the past decade, the rapid development of whole-genome sequencing (WGS) has enabled more detailed evaluation of the intra-host evolution of viral genomes [12,13]. Here, we report on the intra-host evolution of B19V during relapsing infection in three immunocompromised patients. We determined genome-wide mutations in the B19V genome from the plasma of these patients, and we investigate in silico the potential conformational changes in the mutated viral proteins using novel machine learning-based modeling. Additionally, we provide a literature overview on intra-host and inter-host B19V genome evolution.

## 2. Materials and Methods

Patients: All patients with a known relapsing or prolonged B19V infection were selected from the laboratory records of the Leiden University Medical Centre (LUMC) in the period 2011–2019. For each patient, four to six B19V DNA-positive serial samples from the laboratory archives were selected for WGS. Samples had previously been sent to the Clinical Microbiology Laboratory (CML) as part of routine patient care for B19V diagnostics, at the discretion of the treating physician. Plasma samples were stored at −80 °C until WGS analysis. Clinical information on clinical background, disease course, laboratory parameters, and treatment was retrospectively retracted from patient records.

Serology: Anti-B19 VP2 IgM and IgG antibodies were measured with the LIAISON^®^ Biotrin Parvovirus B19 assay (DiaSorin, ‘s Hertogenbosch, The Netherlands).

B19V PCR and whole-genome sequencing: Viral loads were determined by quantitative PCR, as previously described [14]. Whole-genome sequencing was performed using the Arc Bio Galileo Pathogen Solution kit, a complete kit and protocol for quantitative metagenomic detection of several DNA viruses in blood of immunocompromised patients, as previously described [15]. In short, patient samples were spiked with an internal baculovirus control before extraction. Nucleic acids were extracted from plasma using the DNA and Viral NA small-volume extraction kit on the MagNAPure 96 system (Roche diagnostics, Almere, The Netherlands). After concentration, library preparation was performed with the Galileo Viral Panel sequencing kit (Arc Bio (present: Cantata Bio), LLC, Cambridge, MA, USA), according to the manufacturer’s instructions. Samples were sequenced using the NovaSeq 6000 platform (Illumina, San Diego, CA, USA) at GenomeScan B. V. (Leiden, The Netherlands).

Bioinformatic analysis: Sequence reads were demultiplexed using bcl2fastq (version 2.2.0) (Illumina, San Diego, CA, USA), resulting in FASTQ files. De novo assembly was performed using SPAdes (version 3.11.1). Contigs were mapped against B19V reference genome NC_000883.2. The threshold for nucleotide consensus was set at >50% to select mutations that were dominant at least at one point in time. Subsequently, the frequency of these resulting mutations was tracked in all samples across all time points. FASTA files were uploaded in Geneious version 2024.0.3 for further comparative and phylogenetic analysis.

Phylogenetic analysis: All available (near-) complete genome sequences (4800 to 5596 bp) of taxonomy ID 10798 (‘parvovirus B19’) and taxonomy ID 344889 (‘unclassified erythrovirus’) were downloaded from the NCBI database. Clonal and artificial sequences were excluded. Alignments were created with Geneious (version 2024.0.3). A Neighbor-Joining phylogenetic tree (Jukes-Cantor model) was constructed. A second tree was constructed using the NS1-VP1/VP2 section of the genomes (≥4280 bp) in relation to European GenBank genotype 1 submissions.

Protein structural modeling: Protein sequences with and without the mutations were subjected to protein structure prediction using Alphafold 2, as well as Alphafold 3 [16]. Mol* was used to visualize the predicted structures and map the predicted local distance difference test (pLDDT) values as heat colors on the structure [17]. pLDDT is a per-residue measure of local confidence as predicted by AlphaFold, with higher scores indicating a more accurate prediction. In addition to the structure predicted by AlphaFold, the computed pLDDT values reflect the molecular dynamic of the predicted structure [18]. Default parameters in AlphaFold were used after updating all required databases as of June 2023. The evaluation focused on the relaxed models. To map the variant protein structures on the capsid, capsid structures of B19V were downloaded from the RCBS Protein Data Bank (PDB) [19,20]; Mol* was used to visualize the predicted structures [17].

## 3. Results

In the period 2011–2019, three patients were identified with relapsing B19V infection. The clinical background; course of viremia; hemoglobulin levels; and immune parameters, including the presence of antibodies and lymphocyte counts, are shown in Figure 1. Patients A and B received a kidney transplant at t = 0 weeks, at which time both patients tested negative for B19V viremia. Patient C received stem cell transplantation (SCT) for Non-Hodgkin lymphoma (NHL) 5 months previously, and t = 0 represents the latest time-point before B19V viremia. During most of the sampling period, patients A and B were lymphopenic due to immunosuppressive treatment. While stable donor chimerism was initially achieved after SCT in patient C, lymphocyte function can be considered progressively impaired due to underlying progressive Non-Hodgkin lymphoma (NHL) after SCT from t = 0 onwards. This patient received donor lymphocyte infusions (DLIs) as part of NHL treatment during the sample period. Patient A and B were B19V-seronegative before transplantation. Patient C was B19V-seropositive before transplantation, but the SCT donor was B19V-seronegative. Patient A first became IgM-positive, then IgG-positive, from 3 weeks after transplantation, but IgG reversion occurred between 20 and 24 weeks after transplantation. Patient B became IgG-positive after IVIG treatment at 12 weeks after transplantation. Patient C first became IgM-positive, then IgG-positive, from t = 7 weeks.

In patients B and C, viral genomes remained unaltered over a course of 3 months and 6 months of follow-up, respectively: no (non-)synonymous mutations were detected by whole-genome sequencing. In patient A, several mutations were identified over a course of 3 months. The first mutation, C3742G, occurred at 14 weeks of follow-up and remained present at 16 weeks of follow-up, while B19V IgG was positive and was no longer detected in the sample at 20 weeks of follow-up when B19V IgG was negative (Figure 1). At 20 weeks of follow-up, mutation A3982T was detected. From this time point onwards, IVIG was administered, and the patient started clearing the infection; no samples with sufficiently high loads were available for sequencing. These mutations were not detected at any time point in patients B and C. Table 1 shows an overview of the detected mutations and the corresponding amino acid changes in patient A.

To assess the impact of amino acid changes on the structure of the virus, protein structural modeling was performed on the mutated VP1 and VP2 proteins of B19V as detected in patient A. The C3742G mutation (patient A) translates into a T145S substitution in VP2, which had no predicted impact on its structure. Likewise, the A3892T mutation resulting in a Q422L/Q195L substitution was predicted not to impact VP1/VP2 structure. Notably, the C3742G mutation and associated amino acid substitution T372S resulted in a predicted conformational change in VP1, while modeling indicated no effect of the same mutation on VP2 conformation. Figure 2 shows the modeled protein structure of the baseline VP1/2 and the modeled protein structure with the amino acid substitution T372S. The T372S structure shows an additional conformational loop in VP1. The high pLDDT values resulting in the loop indicate that the T372S structure has changed from a static to a dynamic structure. The predicted additional loop appears adjacent to the primary antibody binding site (VP1u). Mapping of this variant onto the capsid structure showed that the affected region is part of the exposed capsid surface (Figure 3). Additional mapping onto the known capsid structure with bound human Fab molecules showed that the variant borders the site where the heavy chain of the human antibody binds (Figure 3).

To assess similarity or divergence in relation to other circulating viral strains, we also constructed a phylogenetic tree using (near) whole-genome sequences from GenBank for other published B19V genotype 1–3 viruses (Figure 4). The viruses from patient A (2019) and B (2015) clustered with genotype 1a, while the virus from patient C (2011) clustered with genotype 3a. The virus from patient A clustered most closely with relatively recent strains from Serbia (2011) and other recent viruses from France (2017). The virus from patient B was relatively distant from patient A but still clustered with genotype 1a. Patient C was infected with genotype 3, which is uncommon in Europe, and it is unclear how this patient got infected with a genotype 3 strain. To assess the relation to other local strains, a separate tree was constructed using European GenBank sequences, analyzing a subregion of the genome (4280 bp), which includes the complete NS1-VP1/2 fragments. The viruses from patient A and B clustered closely with other national strains within genotype 1a (Appendix A). As there were very few European GenBank entries for genotype 3, patient C is not included in this analysis.

## 4. Discussion

In this study, we investigated the intra-host evolution of B19V in immunocompromised patients with relapsing B19V infection. In two patients, the viral genome remained stable over several months, despite multiple episodes of intense replication and treatment with IVIG during the sample period in one of the patients. In one patient, two non-synonymous/missense mutations were detected in the VP1/VP2 region, resulting in AA substitutions in four locations. These mutations have not previously been described in the literature (Table 1) and were not present in GenBank sequences used in our study. Modeling of protein folding predicted that one of these mutations (C3742G) would result in a conformational change in VP1.

The B19V capsid is composed of VP1 (5%) and VP2 (95%), while the NS1 protein is not expressed on the surface. Therefore, VP1 and VP2 are considered the most antigenic, which has also been shown in immunization assays [21,22]. The main antigenic domain is thought to be the ‘VP1 unique region’ (VP1u), a protein structure consisting of 227 amino acids on the B19V capsid involved in receptor binding (see Figure 2 and Appendix A) [23]. Thus, the capsid proteins, and VP1u in particular, are the main targets for antibody neutralization. As IVIG is administered in finite doses, as opposed to natural antibody production, and replication flares after IVIG has been cleared from the circulation, we hypothesized that IVIG may induce an evolutionary bottleneck. We hypothesized that humoral immune pressure, and specifically IVIG, might be drivers for B19V evolution, and we expected that changes would most likely occur in the VP1u region. Although we did observe mutations under humoral immune pressure, these mutations did not occur in the region coding for VP1u. Also, modeling of protein folding predicted a conformational change in VP1 but not at the site of VP1u (Figure 2). Although VP1u is considered to be the most antigenic, other (conformational) sites have been shown to be antigenic also. A second important site is the VP1-VP2 junction region [21], although the T372S substitution is not located in this domain. However, it has also been observed that a number of neutralizing antibodies did not react with either capsid proteins or polypeptide fragments, stressing the possibility of conformational epitopes [24]. The conformational change in VP1 we observed indicates a change from a static to a dynamic structure. Mapping of the dynamic region onto the resolved structure of the B19V capsid showed that the affected region of VP1 is a critical part of the capsid surface and borders the site where the heavy chain of the human antibody binds. This suggests a possible role of the predicted conformational change in humoral immune evasion, although additional crystallographic structural studies are needed to confirm the impact on the capsid structure. Subsequently, molecular docking studies, in-depth molecular dynamic simulations, and in vitro binding studies would be needed to fully understand the impact of the predicted molecular changes on, e.g., antibody binding.

In patient A, the first mutations emerged when the patient was B19V IgG-seropositive. The serological profile of patient A indicated a pattern most compatible with natural immunity: the appearance (and disappearance) of IgM, followed by prolonged detection of strong positive signals of IgG. Remarkably, IgG seroreversion occurred shortly after the appearance of mutations (Figure 1). The patient had not yet received IVIG or other blood products that may explain transient IgG-positivity; so, most likely, this indicates a loss of natural immunity, which may be considered rare. Alternatively, IgG may not have been detected by the serological assay due to the observed mutation. This is, however, less likely, considering the assay is based on a recombinant VP2 antigen; in VP2, we did not detect a structural change. In addition, the viral load peaked at the time of IgG disappearance; this is more compatible with true IgG loss than a diagnostic false-negative IgG finding. In patient B, who was sampled during IVIG treatment, no mutations occurred. Patient C was not sampled during IVIG treatment but did receive DLI during the sampling period. This might have resulted in a decrease in viral load (although the SCT donor was B19V-seronegative) (Figure 1). Notably, this primarily T-cell-mediated treatment did apparently not lead to the selection of variants. These results suggest that natural humoral immune pressure might play a more important role in intra-host evolution than IVIG and T-cell-mediated immunity.

Previous studies on B19V intra-host evolution have shown varying results. Table 2 shows an overview of the available literature on B19V intra-host evolution [8,9,10,11,25]. A limitation of all studies is the limited number of study subjects that are included. Also, different regions of the genome were sequenced in these studies, hampering direct comparisons and definite conclusions. One study on intra-host evolution found a relatively high mutation rate compared to other studies [9]. In this study by Hung et al., B19V longstanding infection was studied in three AIDS patients, and a large number of single-nucleotide polymorphisms (SNPs) were observed during a maximum follow-up of 11 months. Remarkably, the large number of mutations only occurred in the two patients treated with highly active antiretroviral therapy (HAART). Only one mutation was found to appear under IVIG treatment. Of note, none of the new strains became dominant. In our study, we observed a disappearance of the nonsynonymous substitution after seroreversion at 18 weeks of follow-up. Combined with the other studies in which relatively few changes in the viral genome were detected in long-term infection and reversion of these changes occurred in the course of follow-up, these studies suggest a relatively high level of genetic stability. As we only found mutations occurring while natural immunity was developing, it could be hypothesized that humoral immune selection is stronger with natural immunity than with IVIG treatment. As we investigated a limited patient population, further studies with larger patient populations would be required to investigate this hypothesis.

Whole-genome phylogenetic analysis on the viruses of our patients showed that our viral sequences clustered with other circulating viral strains, suggesting these are representative strains for these regions and related to other Dutch and European strains. Studies on population evolution fairly consistently report a substitution rate of ~1 × 10^−4^ substitutions per site per year (s/s/y) for B19V, with highest s/s/y reported for the capsid sequences VP1 and VP2 (for an overview of studies, see Appendix A) [8,26,27,28,29,30,31,32]. This is a relatively high substitution rate for DNA viruses, as substitution rates for other DNA viruses are estimated at 10^−5^ to 10^−9^ s/s/y, although with a higher s/s/y for single-stranded DNA viruses (such as B19V) [33]. However, these studies on B19V are based on a relatively short period of observance, which may cause an overestimation of the mutation rate [34]. From an evolutionary point-of-view, B19V intra-host genomic stability is plausible considering the long-term history that B19V and mankind share, as B19V was already demonstrated in 7000-year-old human bone samples [31]. In these studies investigating ancient DNA, substitution rates between 1.02 and 1.22 × 10^−5^ s/s/y were reported [31,32].

## 5. Conclusions

In conclusion, this study showed genomic changes occurring during relapsing B19V infection, which were predicted to lead to a structural capsid protein change. These mutations occurred in the presence of a humoral immune response. Despite the observed relative genetic stability on the population level, much remains unknown about intra-host viral evolution. B19V relapsing infections are a relatively new phenomenon. It is only in the last decades that the circumstances in which B19V may relapse have become present; the development of transplantation medicine has created the existence of consistently severely immunocompromised hosts that may form a reservoir for viral persistence and evolution. In addition, specific treatments such as IVIG may still add to selection pressure, and data on intra-host evolution are still scarce for B19V. We recommend monitoring genome changes and their structural impact on the virus in larger series to increase our understanding of viral evolution under these relatively new circumstances.

## Figures and Tables

**Figure 1 viruses-17-01034-f001:**
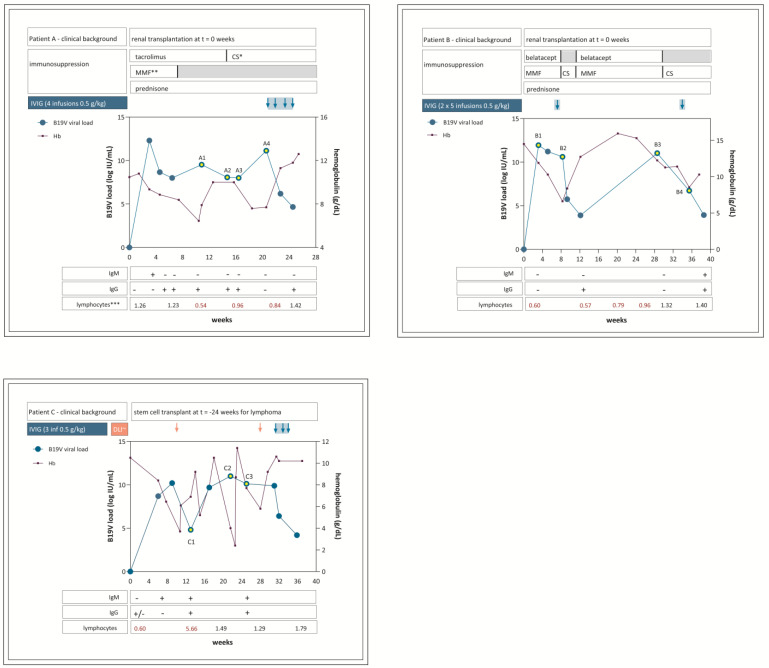
Clinical background, longitudinal course of laboratory parameters, and treatment of 3 patients with relapsing B19V infection. * CS = ciclosporin; ** MMF = mycophenolate mofetil; *** lymphocytes (×10^9^/L), reference values 1–3.5, ~DLI = donor lymphocyte infusion.

**Figure 2 viruses-17-01034-f002:**
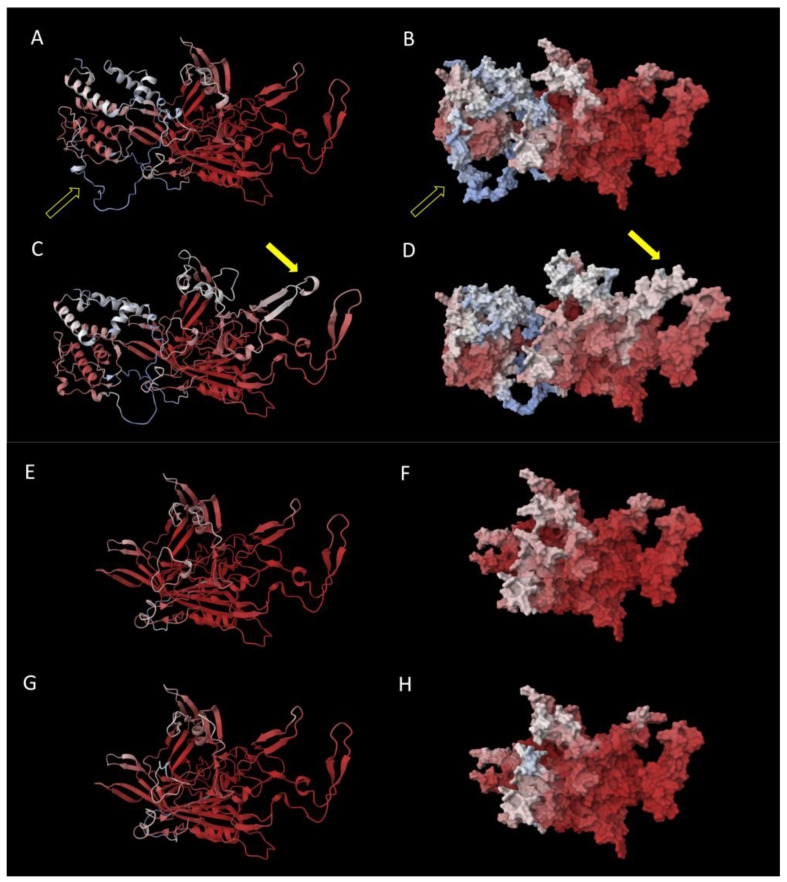
Protein structural modeling of the mutated VP1 (**A**–**D**) and VP2 (**E**–**H**) AA sequence from patient A. (**A**) Baseline protein structure and (**B**) surface representation of VP1, as modeled from the AA sequence for time point A1, baseline (cartoon representation), open arrow indicates VP1u region (for a view of the entire VP1u region, see Appendix A); (**C**) protein structure and (**D**) surface representation of VP1, as modeled from the AA sequence with T372S substitution (time point A2), resulting in an additional loop marked by the closed yellow arrow. (**E**) Baseline protein structure and (**F**) surface presentation of VP2, as modeled from the AA sequence for time point A1 (cartoon representation); (**G**) protein structure and (**H**) surface presentation of VP2, as modeled from the AA sequence with T372S substitution (time point A2), not resulting in structural changes. The heat colors indicate pLDDT values (see Methods Section); red indicates high values. AlphaFold 2 was used to generate the models here, with an analogous analysis using AlphaFold 3 generating similar changes in the 3D structure, which are included in Appendix A.

**Figure 3 viruses-17-01034-f003:**
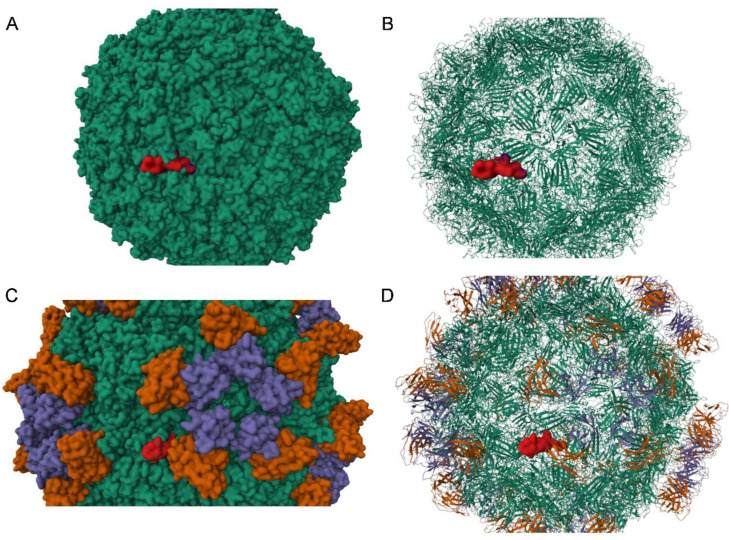
VP1 variable region associated with the studied variant projected onto known capsid 3D structures. In (**A**,**B**), the variable region in red is projected to the structure of the B19V capsid in green (PDB-ID 1S58). (**C**,**D**) show the projection of the variable region in red to the B19V capsid (PDB-ID 6NN3) in green decorated with Fab molecules from a human antibody; heavy chain in orange and light chain in purple. Only one variable region of VP1 in the capsid structure is highlighted.

**Figure 4 viruses-17-01034-f004:**
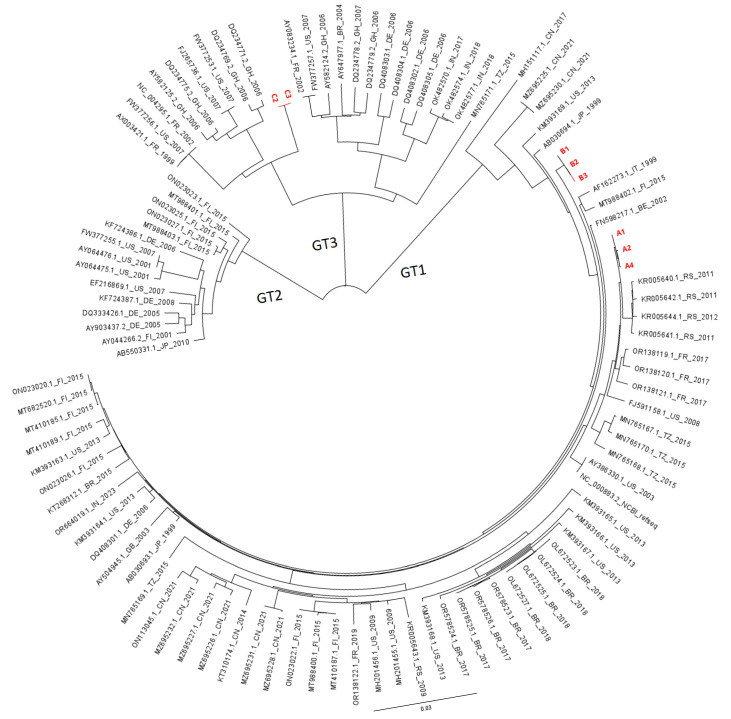
Phylogenetic analysis of patients A and B sequences and global B19V GenBank strains. Selected Genbank strains > 4800 bp of genotype 1, 2, and 3. GenBank strains are denoted by GenBank entries, followed by country (2-digit code according to ISO-3166-1 alpha 2 country codes) and year of isolation.

**Table 1 viruses-17-01034-t001:** Overview of nucleotide substitutions and associated amino acid substitutions from a relapsing infection in patient A.

		Presence C3742G (% Reads)	Presence A3892T (% Reads)	AA Substitution	AA Substitution	AA Substitution
				VP1	VP2	NS
Patient A	0 weeks					
	10 weeks	20%	2%	-	-	-
	14 weeks	48%	16%	T372S	T145S	na ^1^
	16 weeks	51%	38%	T372S	T145S	na
	20 weeks	27%	71%	Q422L	Q195L	na

^1^ na = not applicable; no synonymous Nt substitutions were detected in patient A. (whole genome).

**Table 2 viruses-17-01034-t002:** Literature overview of studies on intra-host evolution of B19V.

First Author	Year	Study Population	B19V Treatment	Sequencing Method	Nucleotide Region Sequenced	Follow-Up Period	Results
Gallinella	1996	1 patient; chronic anemia	Not mentioned	Sanger	2400–3400	16 months	No changes in viral genome
Plentz	2004	1 patient; bone marrow transplantation	IVIG	Not mentioned	Not mentioned	8 months	3 lasting changes after temporary variations: T3463C C4852G T4867C
Hung	2006	3 patients; AIDS	IVIG (*n* = 3); HAART (*n* = 2)	Sanger	436–2431; 3125–4283	11 months	Pt 1: A3271C Pt 2: 18 SNP (14 N ^1^) Pt 3: 15 SNP (9 N)
Suzuki	2014	1 pediatric patient; cord blood transplantation	IVIG	Sanger	602–5014	29 months	6 SNP T/C941T T/C1037T A1048A/G (N) T1112C T1118T/C A1266A/G (N)
Jain	2018	13 pediatric hematological malignancy patients followed up; 3 patients with at least one mutation detected	Not mentioned	Sanger	1747–2691	6 months	Pt 1: G579A Pt 2: C577G Pt 3: C672G

^1^ N = nonsynonymous mutation.

## Data Availability

Data are available from the corresponding author upon reasonable request.

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
