# Peer review of "Intra-Host Evolution During Relapsing Parvovirus B19 Infection in Immunocompromised Patients"

_viruses, 2025, doi:10.3390/v17081034_

Round 1
Reviewer 1 Report (New Reviewer)
Comments and Suggestions for Authors
Abstract: Be more specific than Backgroung
INTRODUCTION
In the introduction we ask for more detail, how ‘pure red cell aplasia’ can develop.
With respect to molecular mechanisms of mutation, the selective pressures exerted by IVIG to favor variants could be added.
With respect to the techniques used, such as the Sanger technique, which focuses only on portions of the genome, then it could be specified that there is an insufficiency of data already in the laboratory. Along with the objective of the study, you are asked to mention, even briefly for diagnostics, clinical monitoring, or therapy
DISCUSSION:
Regarding the connection between mutation and natural selection: please be more specific about the link between the explanation between mutations and immune pressure.
Delve into the conformational change of VP1 protein.
Line 262: A few studies are cited, but a better explanation of the methodological and clinical differences that might explain the discrepancies found is requested.
Future perspectives are too general: try to be more detailed.
Figure:
Fix Table 1 and 2.
Author Response
Please see the attachment

Reviewer 2 Report (New Reviewer)
Comments and Suggestions for Authors
Review of Manuscript “Intra-host evolution during relapsing parvovirus B19 infection in immunocompromised patients” by Anne Russcher et al..
The authors addressed the intra-host evolution of B19V in three immunocompromised patients with relapsing B19V infections. In two patients, the viral genome remained stable over the complete observation period of 40 weeks, which included episodes of high B19V replication and IVIG treatment. In the third patient, two missense mutations in the VP gene were identified, which affected the amino acids (AA) sequence of both B19V capsid proteins, the major capsid protein VP2 and the minor capsid protein VP1. In in silico analysis, only the T372S AA exchange in VP1, but not the corresponding exchange in VP2, led to a conformational change of the protein.
The study mainly confirms the relatively high genetic stability of B19V already found in previous studies. The approach of whole genome sequencing (WGS) does not seem to provide much additional insights as compared to conventional sequencing of specific regions such as the VP gene alone (see also major point 3). Increasing the power of such studies by including more patients (which admittedly can be a difficult task), may be more beneficial in this regard. The authors speculate, that the conformational change predicted for the VP1 T372S AA exchange mutant may lead to a reduced binding of antibodies. This possibility should be addressed experimentally to further analyze the relevance of this point mutation, which has not described before. Some additional major and minor issues listed in detail below should also be addressed in a revised version of the manuscript.
Major points:
1) Formal issue: For reasons unknown to me, the manuscript was made available in proofreading mode with changes highlighted in color, which makes reading much more difficult.
2) General issue: The resolution of the figures is too low to examine details at higher magnifications.
3) The benefits of employing whole genome sequencing (WGS) as a complex approach for identifying point mutations that may be relevant for immune escape are not immediately apparent. Since the antigenic properties of B19V are determined by linear and conformational epitopes in the capsid, sequencing of the VP1/VP2 regions should be sufficient.
Minor points:
1) In the materials and methods section, it is stated (line 100) that threshold for nucleotide consensus was set at > 50%. Furthermore, in line 141 it is stated that the C3742 first occurred at the 14-week time point. However, in table 1, the percentage of reads harboring the mutation is still under 50% (48%). Please explain.
2) “related“ in line 192 should probably read “relation“.
Author Response
Please see the attachment

Reviewer 3 Report (New Reviewer)
Comments and Suggestions for Authors
The current manuscript describes potential intra-host evolution during relapsing B19 infection in immunocompromised patients. The study is well performed and documented. Importantly, the observed variations derived from this study are very interesting and the methodology applied to interpret a potential impact is intriguing. Although, it is likely that the described variations in VP1/VP2 can lead to immune-escape in host, there are – to this reviewer’s opinion – uncertainties which should be clarified before publication:
- The meaning/impact of the described amino acid substitutions in VP1/VP2 remains somewhat unclear. In particular, it remains vague whether there are surface changes in the capsid, might have an impact on the general structure of the capsid (indirect surface changes) or whether they are part of the flexible N-termini. Since the structure of the B19 capsid is solved, an additional (supplementary) figure might help to clarify this issue.
- As mentioned by the authors, receptor interaction of B19V occurs through interaction with VP1u and obviously attracts a major focus also for neutralization. However, additional sites in the capsid might be involved for neutralization escape and apply to the one or other substitution and would be worth discussing.
- The effect of T372S substitution on the VP1, but not on the VP2 structure is very intriguing and could indeed have a major impact as implied by alpha-fold modeling, particularly regarding neutralization escape. Since the authors clearly focus on assembled capsid rather than individual proteins it should be clarified, whether the modeling applies/reflects the structure determined from VLPs or single proteins. This is of particular importance regarding VP1 for which the C-terminal region takes part in the assembled capsid. Again, it would be helpful to indicate the stretch of the protein of which the structure has been solved, the flexible regions, and the sites of the amino acid changes.
Round 2
Reviewer 2 Report (New Reviewer)
Comments and Suggestions for Authors
Review of revised version of manuscript “Intra-host evolution during relapsing parvovirus B19 infection in immunocompromised patients” by Anne Russcher et al..
In the revised version of the manuscript and the corresponding cover letter the authors have addressed the issues raised in my review of the original manuscript in a detailed and satisfactory fashion. Especially the newly added data in figure 3 projecting the predicted structure of the VP1 T1372S amino acid exchange onto the published B19V capsid structure further supports the hypothesis of the authors that this mutation may play a role in immune escape from neutralizing antibodies.
Reviewer 3 Report (New Reviewer)
Comments and Suggestions for Authors
Thanks to the authors. The manuscript significantly improved and should be published in the present form.
This manuscript is a resubmission of an earlier submission. The following is a list of the peer review reports and author responses from that submission.
Round 1
Reviewer 1 Report
Comments and Suggestions for Authors
This is an interesting, well-written manuscript in which the authors report on the dynamics of genomic mutations and subsequent protein changes during relapsing parvovirus B19 (B19V) infection. The manuscript and supplements contain relevant illustrative information facilitating understanding of the study.
Some minor comments:
Lines 17, 71 and 111 - there is a discrepancy in the periods of the study indicated in the abstract (2011-1019), materials and methods, and results sections (2011-2020). Which one is correct?
Line 142 (Figure 1) - it would be desirable to add a DLI explanation before abbreviation: DLI = donor lymphocyte infusion
Line 225 – ‘donor lymphocyte infusion (DLI)’
Line 236 – ‘highly active antiretroviral therapy (HAART)’
'Tables' and 'Figures' should be capitalised throughout the text (pages 6-8).
Reviewer 2 Report
Comments and Suggestions for Authors
The paper entitled "Intra-Host Evolution During Relapsing Parvovirus B19 Infection in Immunocompromised Patients", submitted by Russcher and colleagues for consideration for publication in Viruses, addresses the very important topic of B19V intrahost evolution.
The authors investigated the B19V genome and related structural evolution in the course of prolonged infections in three immunocompromised patients over 28-40 weeks. Whole genome sequencing and bioinformatic analysis revealed the accumulation of two -possibly consecutive- non-synonymous mutations in one patient; phylogenetic analysis assigned two isolates to genotype 1a, the third being genotype 3. Sequence variations were used as a basis for a structural prediction using Alphafold 2, with an inference on a conformational modification and its possible consequences on immune system recognition.
Overall, the topic might be of interest to the Scientific community, and I read it with great expectations. Unfortunately, several drawbacks limit its scientific soundness. Such issues span from manuscript crafting to technical approaches, data presentation, and interpretation. Therefore, I do not believe that a revision could be performed in the limited time usually offered by this Journal for Revisions, and I have to recommend rejection. My main concerns are as follows:
* The Methods section appears to lack very important details, which would be required for the proper evaluation of the manuscript. For example: how were IgM and IgG measured from patients? This is important because the "seroconversion" in Patient A (i.e. disappearance of IgG vs VP1/2 might be interpreted as the development of IgG against the mutated VP1/2, which cannot recognize the wild-type epitopes. Further experimental details must be provided.
* NGS. It appears the authors performed NGS on clinical samples at different times post-transplant. However, loos like they are considering nucleotide consensus (set at > 50%). This choice is puzzling to me since it necessarily results in the underrepresentation of viral evolutions the authors are attempting to study. The evidence that a second viral mutant arises in the course of infection, and which is wild-type for the site mutated originally, further demonstrated that the patient has different viral genomes co-existing during infection. However, whether they have been generated during infection or they initially infected patient A, is not known. A more detailed analysis of the viral population circulating in Patient A during infection is required. In other words, the manuscript does not provide convincing evidence for de novo generation of viral mutants and lacks sufficient resolution.
3) Predictions. The authors proposed that the mutations cause a conformational change in VP1/2, which is believed to confer immune escape properties. Such a hypothesis is intriguing but not corroborated by experimental data. Predictions with AlphaFold2 are now outdated (AlphFold3 has been around for a while now), and neutralization assays with patients' antisera would be required to strengthen the manuscript.
4) Phylogenesis. It is not immediately clear what the utility of the phylogenetic analysis provided in Figure 3 is.
Given the above-mentioned issues, very little reliable information is contained in this manuscript. However, I feel that an accurate revision taking into account the above points would result in a very interesting study, and I therefore would encourage the authors to resubmit their work once they have addressed my concerns.